# Multimodal Taylor Series Network For Misinformation Detection

Authors

## Abstract

With the rapid development of the Internet and the widespread use of social media, the proliferation of multimodal misinformation combining images and text poses serious risks to societal trust, individual well-being, and the integrity of AI models trained on such data. Recently, the automatic detection multimodal misinformation has become an essential area of research. However, traditional methods often rely on hierarchical neural networks that compress and fuse modalities, potentially overlooking deeper interactions between modalities and reducing model interpretability. In this paper, we present a novel Multimodal Taylor Series (MTS) network for detecting multimodal misinformation. The MTS network leverages Taylor series expansion to explicitly capture both low-order and high-order interactions between modalities, which also enhances interpretability by decomposing the model's processing into distinct terms. Additionally, the proposed MTS network avoids exponential parameter growth and maintains linear scalability, allowing the model to effectively capture complex cross-modal correlations. Extensive experiments on three benchmark datasets demonstrate that the MTS network significantly outperforms state-of-the-art models. We will release our code after the final publication of the paper.

## CCS Concepts

• **Computing methodologies** → **Machine learning**.

## Keywords

Multimodal, Misinformation Detection, Taylor Series Network

## 1 Introduction

With the rapid growth of the internet and social media, the proliferation of multimodal content combining images and text has significantly facilitated the spread of misinformation, posing substantial risks to both societal stability and individual well-being [14, 30]. In the era of large language models, the use of harmful information such as misinformation in training data can result in biased and erroneous outputs, including hallucinations [11], severely misleading users. Consequently, the detection of multimodal misinformation have become critically important.

Traditional approaches [4, 15, 19, 33] to multimodal misinformation detection typically employ hierarchical neural networks that progressively compress multimodal data. These methods often process image and text features separately, followed by modality fusion through simple feature concatenation or attention mechanisms. However, they frequently struggle to effectively capture the intricate, deep interactions between modalities [1]. Furthermore, the data flow in such methods is implicit, making it unclear which specific parameters are responsible for capturing relevant features, thus limiting model interpretability.

To address these limitations, we propose the Multimodal Taylor Series (MTS) network, which effectively handles the inherent complexity of multimodal data. Our approach treats image and text modalities as two variables and leverages a bivariate Taylor series

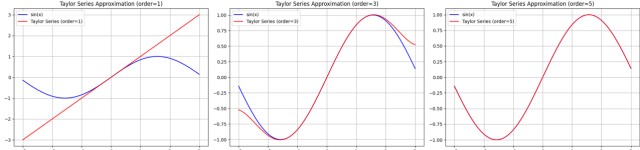

**Figure 1: An illustration of the impact of gradually increasing the order of the Taylor series on function approximation accuracy.**

expansion to approximate the mapping from network input to output. As shown in Figure 1, using the sine function as an example, the approximation becomes increasingly accurate as the order of the Taylor series increases, capturing finer details of the function.

Utilizing a bivariate Taylor series expansion for multimodal misinformation detection offers two key advantages. First, the Taylor series, with its different orders, can explicitly model both shallow and deep interactions between image and text modalities, from low-order to high-order interactions. Second, it decomposes the data flow into distinct terms, such as modality-independent terms and cross-modal interaction terms, enabling clear identification of the roles of different parameters and thereby enhancing model interpretability.

To prevent the exponential growth of parameters typically associated with increasing the order of the Taylor series, we simplify the expansion by applying modality-wide partial derivations of the encoded features and generating expansion terms in an efficient way. This optimization improves both the scalability and practicality of the model, making it easy to implement and deploy in real-world applications. In the experiments, we evaluate the proposed method on benchmark datasets for fake news and sarcasm detection. The results consistently show that the MTS network outperforms state-of-the-art methods. Additionally, we also provide a detailed analysis demonstrating the interpretability of the model.

## 2 Related Work

### 2.1 Misinformation Detection

With the widespread use of the Internet and social media, multimodal misinformation detection has recently emerged as an important research area. For instance, CAFE [4] combines unimodal features with cross-modal correlations by applying cross-modal fuzzy learning and modal fusion. FND-CLIP [33] employs CLIP to perform normalized and weighted fusion, reducing redundant information in both image and text features. LogicDM [19] leverages fine-grained word-level embeddings processed by an LSTM, incorporating additional predicate features to enhance interpretability and performance. BMR [29] adapts image and text features using consistency learning, followed by feature extraction and fusion through a multi-expert network. FSRU [15] improves rumor detection by integrating spectral features from both modalities. These methods are typically based on deep neural networks.

In contrast, this paper introduces a novel network architecture based on Taylor series expansion, which effectively captures interactions between modalities at various orders while offering clearer model interpretability.

## 2.2 Polynomial Networks

Research has demonstrated that polynomial methods can achieve superior fitting with fewer parameters, even without activation functions [5, 8, 20]. For example, Π-Nets [5], a new class of deep convolutional neural networks (DCNNs), utilize polynomial neural networks for function approximation. These networks employ unique jump connections to form higher-order polynomials, extending traditional compositional paradigms. Similarly, CAT [8] uses a concept encoder alongside a polynomial network based on Taylor expansions, delivering strong performance across various single-modal tasks.

However, most prior work has focused on single-modal scenarios, limiting its applicability to multimodal tasks. One major limitation of previous methods is the exponential growth in the number of parameters with increasing polynomial degree, making it impractical to model higher-order complex functions. Polynomial structures of only 2-3 degrees struggle to capture intricate distributions of multimodal data and deep interactions between modalities. In contrast, our method is specifically designed for multimodal scenarios, ensuring that parameter growth remains linear with increasing polynomial degree. This addresses the limitations of earlier methods, which suffer from reduced fitting capacity due to exponential parameter growth.

## 3 Methodology

### 3.1 Overview

The overall framework is depicted in Figure 2. Given a multimodal input sample, we first encode the image and text modalities into their respective feature representations, $e_i$ for the image and $e_t$ for the text. For the image modality, we utilize ResNet followed by a linear layer to extract the encoded feature. For the text modality, we input the text into BERT, and extract the CLS token, which is then passed through a linear layer to obtain the encoded text feature. These encoded features are then fed into the proposed Multimodal Taylor Series (MTS) network to generate the final feature representations, denoted as $M_n$ and $N_n$. Subsequently, we perform an element-wise addition of the two final feature representations, and pass the result through a linear layer to produce the logits for final classification.

The MTS network is derived from the Taylor series expansion and consists of multiple layers. As the number of layers in the network increases, the overall function represented by the network becomes more expressive, capturing increasingly complex interactions between the image and text modalities. In the next sections, we elaborate how the MTS network is derived from the original Taylor series expansion and adapted for practical implementation. To maintain mathematical clarity, henceforth we use $\vec{x}$ to denote the image feature $e_i$, and $\vec{y}$ to denote the text feature $e_t$.

## 3.2 Multimodal Taylor Series Network

In mathematics, Taylor's theorem states that a polynomial function of $d$ variables can be approximated using a Taylor series expansion, as shown below:

$$f(\vec{x}) \approx \sum_{k=0}^{N} \frac{1}{k!} \left[ \sum_{i=1}^{d} \left( \Delta x_i \frac{\partial}{\partial x_i} \right) \right]^k f \Big|_{x_0}, \tag{1}$$

where $\vec{x} \in \mathbb{R}^d$, and $N$ denotes the order of the Taylor series. To adapt this for multimodal data, we extend the equation to a bivariate form to better represent the interactions across multiple modalities. For simplicity, we perform the Taylor series expansion at the origin:

$$f(\vec{x}, \vec{y}) \approx f(\vec{0}, \vec{0}) + \sum_{k=1}^{N} \frac{1}{k!} \left( \sum_{i=1}^{d} x_i \frac{\partial}{\partial x_i} + \sum_{j=1}^{d} y_j \frac{\partial}{\partial y_j} \right)^k f \Big|_{(\vec{0},\vec{0})}. \tag{2}$$

When incorporating tensors, this can be written as:

$$f(\vec{x}, \vec{y}) \approx f(\vec{0}, \vec{0}) + \sum_{k=1}^{N} \left( W^{[k]} \prod_{i=2}^{k+1} \hat{\times}_i [\vec{x}, \vec{y}]^\top \right), \tag{3}$$

where $\vec{x} \in \mathbb{R}^d$ and $\vec{y} \in \mathbb{R}^d$ are the feature representations of the image and text modalities, respectively, and $[\vec{x}, \vec{y}] \in \mathbb{R}^{2d}$ denotes the concatenation of the two features. $W^{[k]} \in \mathbb{R}^{\prod_{n=2}^{k} \times (2d)}$ represents the $k$-th order derivative parameters of the function, and $\hat{\times}_j$ denotes the mode-$n$ matrix product [13]. The term $\prod_{i=2}^{k+1} \hat{\times}_i$ represents the operation of applying the $k$-th order derivative of the parameter $W^{[k]}$ and performing element-wise multiplication with $[\vec{x}, \vec{y}]$. The number of parameters required to fit the coefficients $W^{[k]}$ in the $N$-th order polynomial is:

$$O(f(\vec{x}, \vec{y})) = \sum_{k=1}^{N} (2d)^k = \frac{2d \times (1 - (2d)^N)}{1 - 2d}, \tag{4}$$

$$O(f(\vec{x}, \vec{y})) = (2d)^N. \tag{5}$$

As shown in Eq. 5, the number of parameters grows exponentially with the order $N$. While this approach is feasible when $N$ is small, it becomes impractical for larger values of $N$, as it requires an excessive number of parameters to capture the complex interactions among different modalities. To address this scalability issue, we propose reducing the parameter size in Eq. 3.

## 3.3 Modality-Wide Partial Derivation of Encoded Features

To mitigate the exponential growth of parameters in Eq. 3 with increasing order $N$, we treat each modality's encoded feature as a whole for partial derivation, rather than treating each dimension of the feature as a separate variable. For example, for the image feature $\vec{x} \in \mathbb{R}^d$, we compute $\frac{\partial}{\partial \vec{x}}$ instead of $\left\{ \frac{\partial}{\partial x_1}, \frac{\partial}{\partial x_2}, \ldots, \frac{\partial}{\partial x_d} \right\}$. As a result, Eq. 3 transforms into:

$$f(\vec{x}, \vec{y}) \approx \sum_{k=0}^{N} \left( W_{\vec{x}}^k \vec{x}^\top + W_{\vec{y}}^k \vec{y}^\top \right)^k, \tag{6}$$

where the Hessian matrices $W_{\vec{x}}^k$ and $W_{\vec{y}}^k \in \mathbb{R}^{d \times d}$ represent the $k$-th order partial derivative matrix. The number of parameters

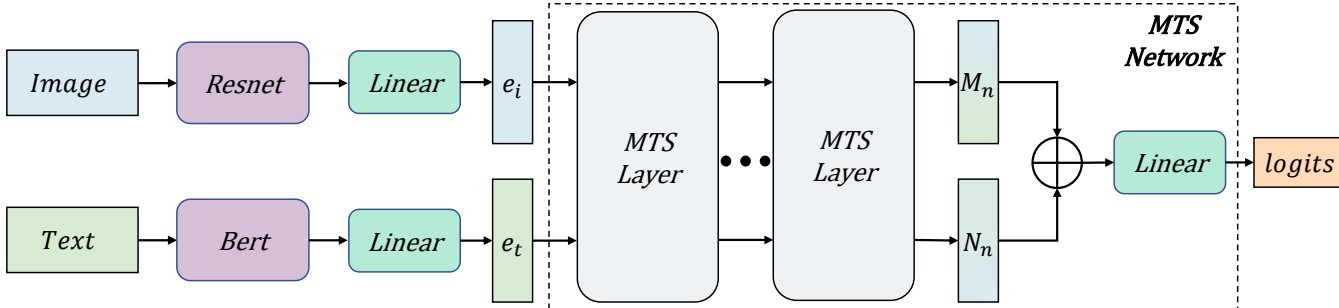

**Figure 2: The overview of the proposed framework. Note that all the linear layers are single-layer linear transformations without activation functions, in order to preserve the original structure and properties of the Taylor series expansion.**

required to fit this improved expression is:

$$O(f(\vec{x}, \vec{y})) = \sum_{k=1}^{N} (2d)^2 = N \times (2d)^2. \tag{7}$$

This reduces the parameter count from $(2d)^N$ to $N \times (2d)^2$, transforming the parameter growth from exponential to linear and significantly improving scalability. As a result, the model is capable of fitting higher-order, more complex functions.

Moreover, treating the modality-wide encoded feature as a whole is well-founded in multimodal learning. The semantic integrity of an encoded image or text feature should be preserved in the form of a complete vector, which is crucial for modality representation learning. If we treat each dimension separately, we risk breaking the semantic integrity of the feature and ignoring the interconnections between different dimensions, thereby weakening the expressive power of the encoded features.

### 3.4 Efficient Generation of Taylor Series Expansion Terms

Although Eq. 6 reduces the number of parameters, directly implementing the network architecture based on this equation remains suboptimal. In Eq. 6, the features of the two modalities are simply linearly added, meaning they are immediately combined into a single representation without preserving the distinct information from each modality. However, in multimodal learning, due to the semantic and modality gaps present in the encoded features of different modalities [17], such an early combination can negatively impact the representation learning of modality-specific features and reduce the model's ability to capture cross-modal interactions effectively.

To address this, one straightforward solution is to transform Eq. 6 into its full expansion form. The expanded expression must include terms of all orders from 0 to $N$, covering both single-modality features and cross-modal combination terms. Omitting coefficient matrices for clarity, the cumulative sum of all terms can be expressed as follows:

$$A^N = \sum_{k=0}^{N} \sum_{i=0}^{k} \left( x^i \times y^{k-i} \right), \tag{8}$$

The relationship between the total number of terms and the order $N$ is given by:

$$O_{\text{term}} (f(\vec{x}, \vec{y})) \approx 2^N. \tag{9}$$

As the order $N$ increases, the number of terms grows exponentially, resulting in a proportional increase in the number of parameters. Consequently, implementing the full expansion of Eq. 6 is impractical. Manually setting all single-modality and cross-modal terms would be cumbersome, and the exponential growth in parameters would severely limit the model's scalability.

To solve this issue, we transform the process of generating all terms in the full expansion of Eq. 6 into a simpler and more efficient process for practical implementation. Specifically, we define two bivariate functions, $\mathcal{M}(\cdot, \cdot)$ and $\mathcal{N}(\cdot, \cdot)$, as follows:

$$\mathcal{M}(a, b) = a + ab + a^2, \quad \mathcal{N}(a, b) = b + ab + b^2, \tag{10}$$

where $a$ and $b$ are two variables. We further define the initial terms $M_0$ and $N_0$ as:

$$M_0 = \mathcal{M}(\vec{x}, \vec{y}), \quad N_0 = \mathcal{N}(\vec{x}, \vec{y}). \tag{11}$$

We can divide $M_0$ (and similarly $N_0$) into three components for finer analysis. The term $\vec{x}$ serves as a linear term that preserves low-order features. The term $\vec{x}\vec{y}$ acts as an interaction term, capturing mutual information between modalities and elevating $\vec{x}$ and $\vec{y}$ to higher-order terms. Lastly, $\vec{x}^2$ functions as a squared term, focusing on high-order feature extraction. We can further simplify this into the following form:

$$M_0 = Vecx(I + \vec{x} + \vec{y}) = \vec{x}A^1, \quad N_0 = \vec{y}(I + \vec{x} + \vec{y}) = \vec{y}A^1, \tag{12}$$

We then define the next terms $M_1$ and $N_1$ based on Eq. 12 as:

$$M_1 = \mathcal{M}(M_0, N_0) = \vec{x}A^1(1 + \vec{x} + \vec{y} + 2\vec{x}\vec{y} + \vec{x}^2 + \vec{y}^2) = \vec{x}A^3, \tag{13}$$

$$N_1 = \mathcal{N}(M_0, N_0) = \vec{y}A^1(1 + \vec{x} + \vec{y} + 2\vec{x}\vec{y} + \vec{x}^2 + \vec{y}^2) = \vec{y}A^3. \tag{14}$$

By following this pattern, we generalize the expressions for $M_n$ and $N_n$ as:

$$M_n = \mathcal{M}(M_{n-1}, N_{n-1}) = \vec{x}A^{2^n - 1}(1 + (\vec{x} + \vec{y})A^{2^n - 1}) = \vec{x}A^{2^{n+1} - 1}, \tag{15}$$

$$N_n = \mathcal{N}(M_{n-1}, N_{n-1}) = \vec{y}A^{2^n-1}(1 + (\vec{x}+\vec{y})A^{2^n-1}) = \vec{y}A^{2^{n+1}-1},$$
(16)

$$M_n + N_n = A^{2^{n+1}}.$$
(17)

As shown in Eq. 17, if we set $n = \log_2 N - 1$, we can use $M_n + N_n$ to express all constituent terms in Eq. 8. In our method, we implement this progressive derivation process as a multi-layer network. As shown in Figure 3, each layer corresponds to the iteration from $M_{n-1}$ to $M_n$. The original input to $M_0$ and $N_0$ are the encoded image and text features. This network structure allows us to express all constituent terms in Eq. 8 with only logarithmic growth in the number of terms, significantly reducing the number of parameters and computational complexity while preserving the ability to capture complex multimodal interactions.

In practice, each term is preceded by a corresponding coefficient matrix (omitted in Eq. 8 for clarity). The final expression for $M_n$ (and similarly for $N_n$) is given in Eq. 18. The full formulation of the proposed multimodal Taylor series network is presented in Eq. 19.

$$M_n = W_\gamma M_{n-1} + (W_{\alpha_1} M_{n-1}) \odot (W_{\alpha_2} N_{n-1}) \\ + (W_{\beta_1} M_{n-1}) \odot (W_{\beta_2} M_{n-1}),$$
(18)

$$f(\vec{x}, \vec{y}) \approx M_n + N_n, \quad \text{where } n = \log_2 N - 1,$$
(19)

where $W_\gamma, W_{\alpha_1}, W_{\alpha_2}, W_{\beta_1}, W_{\beta_2} \in \mathbb{R}^{d\times d}$ are coefficient matrices, and $\odot$ denotes the Hadamard product. From Eq. 19, we see that this approach preserves both low-order and high-order information from different modalities while effectively capturing cross-modal correlations.

To incorporate semantic meaning into the notations $M_n$ and $N_n$, in the experiments, we refer to them as the Text-Guided Image Refinement (TIR) feature and the Image-Guided Text Refinement (ITR) feature, respectively. In TIR, the text modality is dominant, while the image serves as a supervisory signal to adjust inter-modal correlations. Conversely, in ITR, the image modality is dominant and influenced by the text.

## 3.5 Infinity Norm Scaling

To maintain the original structure and properties of the Taylor series, we avoid introducing nonlinear factors such as sigmoid, tanh, or batch normalization for magnitude scaling within the network architecture. However, without any form of scaling on the data flow within the network, this can lead to a magnitude explosion in the network output, as well as gradient explosion during back-propagation. To address this, we propose an infinity norm scaling strategy.

Specifically, for the $n$-th layer of the network, we first aggregate a batch of $M_n \in \mathbb{R}^d$ and $N_n \in \mathbb{R}^d$ into $M_{batch} \in \mathbb{R}^{batch\times d}$ and $N_{batch} \in \mathbb{R}^{batch\times d}$, respectively, where $batch$ represents the actual batch size. We then apply the $L_\infty$ norm constraint to $M_{batch}$ and $N_{batch}$, scaling the values in the batch data for each layer to the range of -1 to 1, as shown in the following formula:

$$M'_{batch} = \frac{M_{batch}}{||M_{batch}||_\infty}, \quad N'_{batch} = \frac{N_{batch}}{||N_{batch}||_\infty},$$
(20)

where $|| \cdot ||_\infty$ denotes the maximum absolute value within the batch. This ensures that the data flow remains stable, preventing magnitude and gradient explosions.

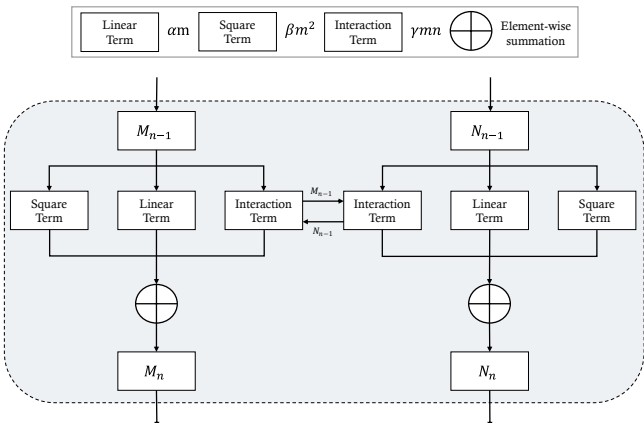

**Figure 3: The architecture of an MTS layer. Here, $m$ and $n$ represent the feature vectors of the two modalities in the current layer, while $\alpha$, $\beta$, and $\gamma$ are learnable parameters.**

## 3.6 Low-Order Independent Initialization

In each layer of the proposed network, the parameters $W_{\alpha_1}, W_{\alpha_2}, W_{\beta_1}, W_{\beta_2}$ for the square and interaction terms in Eq. 18 are initialized using Xavier uniform initialization [9]. For the linear term, we initialize its coefficient matrix $W_\gamma$ with an additional identity matrix, i.e., $I + xavier_{init}$, allowing it to act as a residual term as well.

It is crucial to choose an initialization method that ensures the square and interaction terms are nearly zero at the beginning of network training. Otherwise, it would impose a prior assumption on the higher-order features of each modality and the interaction between the two modalities, hindering comprehensive learning. Furthermore, this would not align with the principle illustrated in Figure 1, where the Taylor series progressively fits complex functions by learning from low-order to high-order terms. Next, we briefly demonstrate why our initialization method ensures this.

Assume the feature distribution for the image modality follows $N_{\text{Img}}(0, 1)$ and for the text modality follows $N_{\text{Text}}(0, 1)$, representing two independent normal distributions. After applying the coefficient matrix, the distributions remain unchanged because the Xavier-initialized matrix ensures that the variance of the features before and after input remains consistent.

After applying infinity norm scaling (Section 3.5), the distribution of the interaction term can be expressed as:

$$\text{Inter} = \frac{N_{\text{Img}}(0, 1)}{\mathbb{E}(\text{Max}[N_{\text{Img}}(X_{1,...,batch})])} \cdot \frac{N_{\text{Text}}(0, 1)}{\mathbb{E}(\text{Max}[N_{\text{Text}}(Y_{1,...,batch})])},$$
(21)

where $X_{1,...,n}$ and $Y_{1,...,n}$ represent a batch of independent samples from the $N_{\text{Img}}$ and $N_{\text{Text}}$ distributions, respectively, and $\mathbb{E}(\text{Max}[\cdot])$ is the expected value of the $L_\infty$ norm.

From a complex mathematical derivation (details omitted), we know that for samples $X_1, \ldots, X_n \sim N(0, 1)$, the approximation of the expected value of the $L_\infty$ norm is:

$$\mathbb{E}(\text{Max}[X_1, \ldots, X_n]) \approx \sqrt{2\log n}.$$
(22)

| Dataset | Weibo | Pheme | Sarcasm |
|---|---|---|---|
| Language | Chinese | English | English |
| Total Samples | 7532 | 2018 | 22225 |
| Misinformation | 3749 | 590 | 9601 |
| Non-Misinformation | 3783 | 1428 | 12624 |

**Table 1: Statistics of the Datasets.**

Thus, the distribution function of the interaction term in Eq. 21 can be transformed as follows:

$$\text{Inter}_{\text{Term}} \approx \frac{N_{\text{Img, Text}}(0, 1)}{2 \log(\text{batch} \times d)}, \tag{23}$$

where $N_{\text{Img, Text}}(0, 1)$ represents a bivariate normal distribution with a mean of 0 and variance of 1, comprising both image and text features. The expected mean of the distribution in Eq. 23 is 0, and the corresponding variance is $\frac{1}{(2 \log(\text{batch} \times d))^2}$.

Given the experimental settings where the batch size is 24 and $d = 768$, the variance is approximately 0.0012, which is very close to 0. This indicates that, after initialization and without any training, the values of the interaction and square terms fluctuate within a narrow range around 0, effectively behaving as if these terms are not included.

In this way, the network retains only the most basic low-order features in the early stages of training. Since the initialization method is applied to all layers, the entire network can initially be viewed as a nearly linear combination of two independent variables, i.e., $f(\vec{x}, \vec{y}) = g(\vec{x}) + h(\vec{y})$. As training progresses, the learning of higher-order coefficients gradually captures the higher-order function distribution. This process also implicitly learns the correlation between the two modalities, achieving modality fusion within the polynomial expression.

## 4 Experiments

### 4.1 Experimental Setup

*4.1.1 Datasets.* We evaluate the proposed method on three benchmark datasets for two tasks: Weibo [24] and Pheme [34] for multimodal fake news detection, and Sarcasm [2] for multimodal sarcasm detection. For Weibo and Sarcasm, we use the partitioning strategy from LogicDM [19], and for Pheme, we follow MFAN's [31] method. Dataset statistics are shown in Table 1. Fake news and sarcasm are treated as misinformation, while real news and non-sarcastic expressions are considered non-misinformation.

*4.1.2 Compared Methods.* We compare the proposed MTS network with three categories of baselines:

**1) Unimodal Models.** These methods operate on a single modality, either images or text. For text-based models, we include BERT [6], while for image-based models, we use ViT [7].

**2) Multimodal Models Using Only Original Samples.** This category includes methods such as EANN [25], MVAE [12], SAFE [32], CAFE [4], SpotFake [23], SpotFake+ [22], FSRU [15], and MM-CAN [10]. These models rely solely on the original image and text content of the samples without incorporating external knowledge or additional input data. Their primary objective is to enhance feature fusion between different modalities, fully leveraging the available information from both image and text modalities.

**3) Multimodal Models Using Additional Information.** This category includes models such as MCNN [28], MCAN [26], BMR [29], LogicDM [19], and MFAN [31]. These methods incorporate additional knowledge or external input data to provide the model with more context and information about the samples, thereby improving the performance of the detection task.

Note that the proposed MTS network only utilizes the original image and text data, without integrating any auxiliary information like the third group of methods. We include these methods for completeness and comparison, as it allows us to demonstrate that even without the use of external knowledge, our approach achieves superior results. This highlights the effectiveness of the proposed MTS network in capturing and utilizing multimodal interactions.

*4.1.3 Implementation Details.* For the text modality, we utilize the BERT-base-chinese model [1] to encode the text data in the Weibo dataset, which is in Chinese. For the English datasets, including Pheme and Sarcasm, we employ the BERT-base-uncased model [2], setting the maximum number of input text tokens to 200. For the image modality, we use ResNet34 [3] as the image encoder. Input images are resized to 224x224 pixels before being fed into the network. The output features from both the image and text encoders are subsequently projected into 768-dimensional vectors using a linear layer, before being input into the proposed MTS network. The model is trained using the cross-entropy loss function, and we optimize it with the AdamW optimizer, setting the learning rate to $2 \times 10^{-5}$. We use a batch size of 24 in all experiments.

*4.1.4 Evaluation Metrics.* To evaluate our method and compare it with baseline models, we use standard performance metrics including accuracy (Acc), precision (P), recall (R), and F1 score (F1).

### 4.2 Comparison Results

**Multimodal Fake News Detection.** As shown in Table 2, the proposed MTS network consistently delivers superior performance across both the Weibo and Pheme datasets, outperforming all baseline methods on the majority of metrics. Compared to baselines that do not utilize external information, MTS demonstrates significant improvements. For instance, compared to the competitive baseline MMCAN-Res, on the Weibo dataset, MTS improves the F1-Score for fake news and real news by 1.6% and 0.9%, respectively. Similarly, on the Pheme dataset, MTS improves the F1-Score for fake news by 4.8% and for real news by 1.7%.

Even when compared to methods that leverage external knowledge, such as MFAN, MTS still outperforms them across most metrics. For example, MTS improves accuracy by 2.8% on Weibo and by 2.6% on Pheme, further underscoring its effectiveness without relying on additional external data.

**Multimodal Sarcasm Detection.** As shown in Table 3, the MTS network significantly outperforms all baseline methods in the sarcasm detection task. Notably, compared to the competitive LogicDM model, MTS improves accuracy by 1.4% and F1-Score by 3.9%, illustrating the model's capacity for handling nuanced multimodal content.

---

[1] https://huggingface.co/bert-base-chinese

[2] https://huggingface.co/bert-base-uncased

[3] https://pytorch.org/vision/main/models/generated/torchvision.models.resnet34

| Dataset | Methods | Accuracy | Fake News | | | Real News | | |
|---|---|---|---|---|---|---|---|---|
| | | | Precision | Recall | F1-Score | Precision | Recall | F1-Score |
| Weibo | EANN [25] | 78.2 | 82.7 | 69.7 | 75.6 | 75.2 | 86.3 | 80.4 |
| | MVAE [12] | 82.4 | 85.4 | 76.9 | 80.9 | 80.2 | 87.5 | 83.7 |
| | SAFE [32] | 76.3 | 83.3 | 65.9 | 73.6 | 71.7 | 86.8 | 78.5 |
| | CAFE [4] | 84.0 | 85.5 | 83.0 | 84.2 | 82.5 | 85.1 | 83.7 |
| | SpotFake [23] | 86.9 | 87.7 | 85.9 | 86.8 | 86.1 | 87.9 | 87.0 |
| | SpotFake+ [22] | 87.0 | 88.7 | 84.9 | 86.8 | 85.5 | 89.2 | 87.3 |
| | FSRU [15] | 90.1 | 92.2 | 89.2 | 90.6 | 87.9 | 91.3 | 89.5 |
| | MMCAN-Res [10] | 90.6 | 91.6 | 89.7 | 90.6 | 89.8 | 91.6 | 90.7 |
| | MCNN* [28] | 84.6 | 80.9 | 85.7 | 83.2 | 87.9 | 83.7 | 85.8 |
| | MCAN* [26] | 87.8 | 90.1 | 86.2 | 88.1 | 87.0 | 90.3 | 88.6 |
| | BMR* [29] | 88.4 | 87.5 | 88.6 | 88.0 | 87.4 | 88.1 | 87.7 |
| | LogicDM* [19] | 85.2 | 86.2 | 84.5 | 85.3 | 84.3 | 85.9 | 85.1 |
| | MFAN* [31] | 89.1 | **94.2** | 83.5 | 88.5 | 85.0 | **94.8** | 89.6 |
| | MTS (Ours) | **91.9** | 93.7 | **90.7** | **92.2** | **90.1** | 93.3 | **91.6** |
| Pheme | EANN [25] | 68.1 | 68.5 | 66.4 | 69.4 | 70.1 | 75.0 | 74.7 |
| | MVAE [12] | 85.2 | 80.6 | 71.9 | 76.0 | 89.1 | 91.7 | 89.3 |
| | SAFE [32] | 81.1 | 82.7 | 55.9 | 66.7 | 80.6 | 94.0 | 86.6 |
| | CAFE [4] | 86.1 | 81.2 | 64.5 | 71.9 | 87.5 | 94.3 | 90.8 |
| | SpotFake [23] | 82.3 | 74.3 | 74.5 | 74.4 | 86.4 | 86.6 | 86.3 |
| | SpotFake+ [22] | 80.0 | 73.0 | 66.8 | 69.7 | 83.2 | 86.9 | 85.0 |
| | MMCAN-Res [10] | 89.0 | 80.3 | 79.4 | 79.9 | 92.2 | 92.6 | 92.4 |
| | MCNN* [28] | 82.4 | 80.9 | 77.9 | 79.5 | 83.9 | 87.0 | 85.4 |
| | MCAN* [26] | 86.7 | 81.9 | 82.1 | 82.0 | 88.7 | 88.5 | 88.6 |
| | MFAN* [31] | 88.8 | 77.1 | **84.6** | 80.7 | **93.9** | 90.5 | 92.2 |
| | MTS (Ours) | **91.4** | **89.2** | 80.5 | **84.7** | 92.2 | **96.0** | **94.1** |

Table 2: Performance (%) on the multimodal fake news detection task across the Weibo and Pheme datasets. Methods marked with * incorporate external knowledge or auxiliary information. The best results for each metric are highlighted in bold, and the second-best results are underlined. Some baselines are excluded from the Pheme comparison due to unavailable results.

| | Model | Acc | P | R | F1 |
|---|---|---|---|---|---|
| Unimodal | Bert [6] | 83.9 | 78.7 | 82.3 | 80.2 |
| | ViT [7] | 67.8 | 57.9 | 70.1 | 63.4 |
| Multimodal | HFM [3] | 83.4 | 76.6 | 84.3 | 80.2 |
| | D&R Net [27] | 84.0 | 78.0 | 83.4 | 80.6 |
| | Att-Bert [21] | 86.1 | 80.9 | 85.1 | 82.9 |
| | InCrossMGs [16] | 86.1 | 81.4 | 84.4 | 82.8 |
| | HCM [18] | 87.4 | 81.8 | 86.5 | 84.1 |
| | LogicDM* [19] | 88.1 | 85.7 | 85.0 | 85.3 |
| | Ours | **89.5** | **89.0** | **89.4** | **89.2** |

Table 3: Performance (%) on multimodal sarcasm detection.

These results strongly demonstrate the effectiveness of the proposed MTS network in capturing deep interactions between different modalities. The consistent improvements across both fake news and sarcasm detection tasks highlight the model's versatility, making it a promising solution for a wide range of multimodal classification problems.

### 4.3 Ablation Study

As shown in Table 4, the ablation study investigates the contributions of different components in the proposed model. First, comparing Row 5 with Row 1, we observe a significant performance drop on the Sarcasm dataset when the interaction term is removed.

This highlights the importance of the interaction term in capturing cross-modal correlations between text and image. Second, comparing Row 5 with Row 2, we find that removing the square term has minimal impact, likely because high-order polynomial information, which the square term captures, can also be learned through the interaction term. Lastly, comparing Row 5 with Rows 3 and 4, we see that the absence of either the ITR or TIR feature leads to reduced performance, demonstrating the necessity of both features in the proposed network for optimal performance.

### 4.4 Analysis of Feature Space Distribution

As shown in Figure 5 (a), we observe a clear trend of decreasing variance as the number of layers increases. Variance in a feature vector typically reflects data dispersion, with higher variance indicating more scattered data points. Initially, the features are dispersed in a high-dimensional space, but as the MTS network deepens and becomes more complex, the features gradually concentrate into a lower-dimensional subspace, eventually converging into a highly-compressed representation for final classification. This behavior is consistent with deep learning models, where features are progressively refined and information becomes more concentrated with each layer.

We also conduct a qualitative analysis of the feature distribution. As illustrated in Figure 4, we visualize the sample features from the

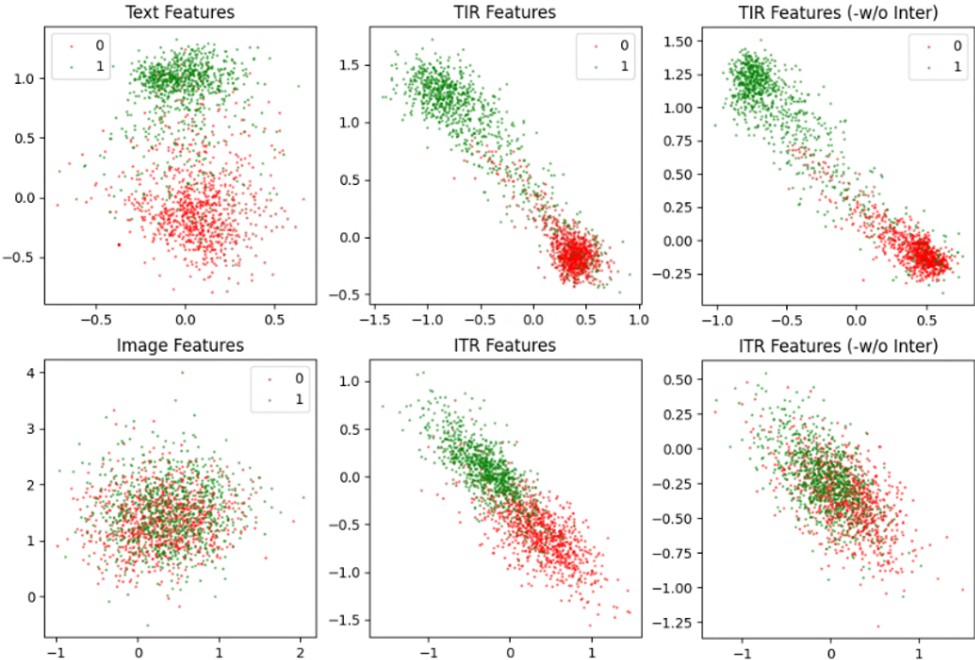

**Figure 4: Feature visualization on a two-dimensional plane. Red and green points represent samples from two different classes. "Image Features" and "Text Features" refer to the original image and text features, $e_i$ and $e_t$, before being input into the MTS network. "ITR Features" and "TIR Features" represent the corresponding output features from the MTS network. The notation "-w/o Inter" indicates that the interaction term has been removed from the features.**

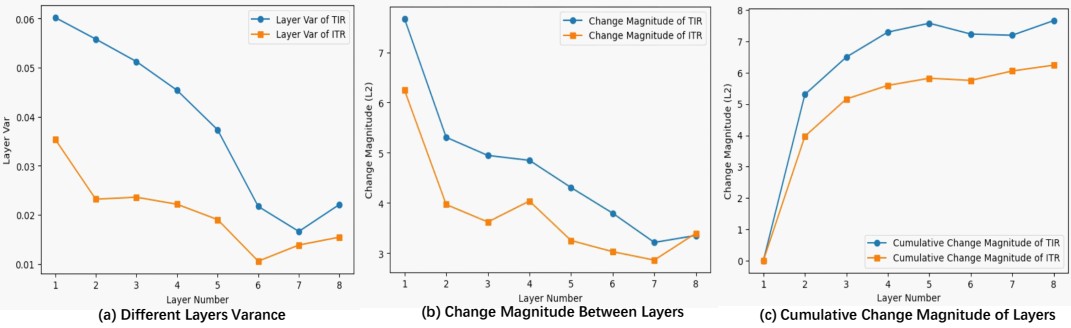

**Figure 5: Change curves of various average values across all samples in the Weibo test set: (a) shows the variance of the ITR and TIR features at each layer of the MTS network; (b) depicts the L2 norm difference of features between consecutive layers, where the vertical value at $l_n$ represents the L2 norm difference between features at layers $l_n$ and $l_{n-1}$; (c) illustrates the cumulative L2 norm change of features from the first layer to the current layer, where the value at $l_n$ represents the cumulative L2 norm difference between features at layer $l_n$ and the first layer.**

Weibo test set by projecting the first two dimensions of each feature onto a 2D plane. Comparing the original features in the first column with the TIR and ITR features processed by the MTS network in the second column, we observe a much clearer separation between the two classes after the MTS processing, highlighting the model's ability to improve classification performance. Furthermore, removing the interaction terms from the ITR features significantly reduces the separability between the classes, underscoring the critical role of interaction terms in capturing cross-modal correlations within

the MTS network. For the TIR features, the difference between retaining and removing the interaction terms is less pronounced. We hypothesize that in the Weibo dataset, the signal from the text modality is already strong enough to dominate the classification, reducing the observable impact of the interaction terms.

| Row | Method | Weibo | | | | Sarcasm | | | |
|-----|--------|----------|-----------|--------|----------|----------|-----------|--------|----------|
| | | Accuracy | Precision | Recall | F1-Score | Accuracy | Precision | Recall | F1-Score |
| 1 | -w/o Inter | 91.5 | 91.6 | 91.3 | 91.4 | 84.6 | 84.1 | 84.0 | 84.0 |
| 2 | -w/o Square | 91.4 | 91.4 | 91.3 | 91.3 | 89.5 | 89.1 | **89.4** | 89.2 |
| 3 | -w/o ITR | 90.7 | 90.7 | 90.7 | 90.7 | 87.6 | 87.0 | 87.4 | 87.2 |
| 4 | -w/o TIR | 89.6 | 90.1 | 89.3 | 89.5 | 89.4 | 89.0 | 89.1 | 89.1 |
| 5 | MTS | **91.5** | **91.6** | **91.4** | **91.5** | **89.5** | **89.1** | 89.2 | **89.2** |

**Table 4: Ablation studies on the different components of the proposed method. "Inter" denotes the interaction term. "Square" denotes the square term.**

| Used Layers | Accuracy | Precision | Recall | F1-Score |
|-------------|----------|-----------|--------|----------|
| None | 62.0 | 62.4 | 62.3 | 61.9 |
| 2 | 88.5 | 88.5 | 88.5 | 88.5 |
| 4 | **91.6** | **91.7** | **91.5** | **91.5** |
| 6 | 91.4 | 91.5 | 91.3 | 91.4 |
| All | 91.5 | 91.6 | 91.4 | 91.5 |

**Table 5: Performance (%) of an 8-layer MTS network on the Weibo dataset using different number of subsets of layers for inference: "None" (no layers), '2', '4', '6', and 'All' (all layers).**

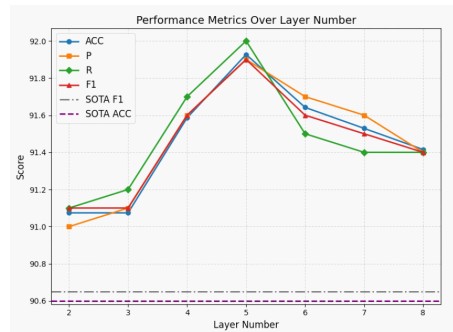

**Figure 6: Performance (%) trend on the Weibo dataset with different layer numbers in the MTS network. The horizontal axis shows the number of layers in the network, and the vertical axis represents the performance metrics. For comparison, we also include the Macro F1-Score and accuracy from the competitive baseline MMCAN-Res, labeled as "SOTA F1" and "SOTA ACC."**

## 4.5 Model Interpretability from the Perspective of Taylor Series

As shown in Figure 5 (b), we observe that the magnitude of feature changes is substantial in the initial layers but diminishes as the model deepens. In Figure 5 (c), the cumulative change increases rapidly at first before stabilizing. These patterns align with the theory of Taylor series expansion, where lower-order terms dominate the capture of core feature interactions, while higher-order terms have diminishing influence and serve primarily as fine-tuners for refining the complex function representation.

To further support this view, we train an 8-layer MTS network and evaluate the impact of using different numbers of layers during testing (see Table 5). The results show that performance improves progressively with the first four layers, while the latter four layers contribute minimally to further enhancement. This observation is also consistent with the Taylor series expansion theory, where lower-order terms are more critical for the model's success, and higher-order terms provide finer adjustments to the overall representation.

## 4.6 Analysis of Layer Number Selection

We investigate the effect of varying layer numbers on the performance of the MTS network, as illustrated in Figure 6, using the Weibo test set. First, we can see that the MTS network consistently outperforms the SOTA baseline in both accuracy and F1-Score, underscoring its robustness across different layer configurations. Second, the network achieves its best performance with a moderate layer count of 5. Performance is notably lower when the layer count is either too small (e.g., 2) or too large (e.g., 8). This suggests that choosing a balanced layer number is essential for optimizing performance and avoiding suboptimal extremes.

In terms of performance trends, we observe that as the number of layers increases from 2 to 5, there is a clear upward trend. This

suggests that higher-order terms in the Taylor series expansion enhance the MTS network's expressiveness, allowing it to model more complex single-modal features and capture deeper cross-modal interactions, leading to improved performance. However, beyond 5 layers, performance begins to decline, indicating that an overly complex Taylor series introduces higher-order terms that overfit the training data, reducing generalization and leading to overfitting.

## 5 Conclusion

In this paper, we propose a novel multimodal Taylor series network for multimodal misinformation detection. Our approach utilizes the polynomial modeling capabilities of the Taylor series to effectively capture the intricate interactions between text and image modalities, while ensuring high interpretability. Additionally, through carefully designed network architecture, we address the challenge of exponential parameter growth and enhance model scalability. Extensive experiments on multiple datasets demonstrate that our method outperforms existing state-of-the-art approaches in both fake news detection and sarcasm detection tasks, even surpassing methods that utilize auxiliary information. Furthermore, our analysis of the different-layer features validate the model's interpretability, while the layer number analysis reveal consistent superiority and robustness. In the future, we hope to extend the proposed method to broader applications in multimodal tasks.

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

Received 20 February 2007; revised 12 March 2009; accepted 5 June 2009