# OpenReview forum: "Multimodal Taylor Series Network For Misinformation Detection"
_ACM.org/TheWebConf/2025/Conference — WWW 2025 Poster_

### Official Review · Reviewer_wA1c · 2024-11-18

**Novelty:** 6
**Technical Quality:** 6

**Review:**

**SUMMARY**:
This paper presents multimodal misinformation detection using Taylor series expansion. The authors propose a Multimodal Taylor Series (MTS) network that explicitly models interactions between image and text modalities. The method demonstrates superior performance on multiple benchmark datasets and provides insights into multimodal feature interactions.

**STRENGTHS**:

1. Strong theoretical foundation
2. Novel approach to multimodal interaction modelling
3. Addresses key limitations of existing methods
4. Comprehensive experimental validation
5. Strong performance improvements over SOTA
6. Thorough ablation studies and analysis
7. Clear practical benefits and implementation details

**WEAKNESSES**: Limited exploration of failure cases

**COMMENTS**:
The paper presents a significant contribution to multimodal learning and web-based misinformation detection. The use of Taylor series expansion for modelling multimodal interactions is novel and well-justified. The authors have done an excellent job addressing practical limitations like parameter growth. The experimental validation is comprehensive, covering multiple datasets and tasks. The ablation studies and analysis also provide some insights into the model's behaviour.

**Questions:**

1. Have you analysed the computational complexity of the model compared to baseline approaches?
2. In section 4.5, what exactly do you mean by "model interpretability"? To me, the analysis seems to be on the surface of exploring behavioural patterns.
- For example, in Figure 5, variance analysis and cumulative change across layers were discussed. While this reveals model behaviour patterns, how does this translate to interpretability of individual decisions?
- If your paper claims interpretability through Taylor series decomposition, how does decomposing the model into terms actually help understand specific predictions? Can you demonstrate with a concrete example how this decomposition helps interpret why the model classified a specific piece of content as misinformation?
- If your paper claims model "interpretability", how does your approach to interpretability compare with other interpretability methods in multimodal learning, such as attention visualisation or feature attribution techniques?

**Reviewer Confidence:**

3: The reviewer is confident but not certain that the evaluation is correct

**Scope:**

3: The work is somewhat relevant to the Web and to the track, and is of narrow interest to a sub-community

---

### Official Review · Reviewer_Zc6L · 2024-11-30

**Novelty:** 4
**Technical Quality:** 4

**Review:**

The authors investigated the issue of misinformation detection and proposed a network architecture based on Taylor series. They claimed that this architecture can more explicitly capture the interaction effects between different modalities, leading to better detection performance compared to existing methods.

The paper is clearly written. The motivation behind the proposed method is easy to follow. Overall, the components of the model design are well-explained. My main concerns lie in the experiments:

* The authors did not conduct a thorough ablation study on the various components of the proposed model architecture. In Section 3.5, norm scaling is used to mitigate gradient explosion issues. What about modern ReLU and its variants? The author should also evaluate the impact of different components on model performance.

* The specific hyperparameters of MTS layer are not clarified.

* I recommend including a comparison of the overhead and model size of the different methods.

* It would be beneficial to provide AUC values for the various methods to better reflect their performance.

* In Table 4, the interaction terms do not appear to yield significant benefits. I suspect that the performance improvement of the proposed method may simply stem from an increase in parameter count.

* What would happen if we concatenate $M_n$ and $D_n$ instead of directly adding them?

* Some baselines are outdated. I suggest that the authors add more recent methods for comparison.

**Questions:**

See above.

**Reviewer Confidence:**

3: The reviewer is confident but not certain that the evaluation is correct

**Scope:**

4: The work is relevant to the Web and to the track, and is of broad interest to the community

---

### Official Review · Reviewer_yz9q · 2024-12-03

**Novelty:** 5
**Technical Quality:** 5

**Review:**

This paper introduces the Multimodal Taylor Series (MTS) Network, an approach for detecting multimodal misinformation by explicitly modeling interactions between text and image data through a Taylor series expansion. Unlike traditional methods that rely on hierarchical neural networks and modality fusion techniques, the MTS network captures both low- and high-order interactions, thereby enhancing interpretability and scalability.

**Strengths:**
+ The paper addresses a critical and growing problem of misinformation, which has profound implications for public perception, politics, and news dissemination.
+ The use of a Taylor series to explicitly model cross-modal interactions is interesting. By decomposing the model's processing into distinct terms, the MTS network provides a higher level of interpretability than conventional approaches. Moreover, the authors successfully address a critical limitation of polynomial models—the exponential growth of parameters—by employing efficient generation of expansion terms and modality-wide partial derivations.
+ The experimental results demonstrate substantial performance improvements over competitive baselines on multiple datasets. This includes both unimodal and multimodal models, as well as approaches that utilize additional external information. The ablation study highlights how specific terms in the Taylor series (e.g., interaction terms and higher-order terms) can be interpreted by humans to gauge modality contributions and the complexity of cross-modal relationships.

**Weaknesses:**
- While the authors demonstrate the effectiveness of their approach on Weibo, Pheme, and Sarcasm datasets, these datasets may not fully represent the diversity of real-world misinformation scenarios. For example, examples sourced from the internet could introduce greater variability in language, cultural contexts, and multimodal data quality. Because the network relies heavily on Taylor series to model cross-modal interactions, its performance on heterogeneous datasets, especially those with significant noise or imbalanced modalities, remains unclear.

- Another area that requires attention is the computational cost of hyperparameter tuning, particularly the determination of the ideal number of layers for the MTS network. The authors provide an analysis of this parameter using the Weibo dataset, showing that five layers yield optimal performance. However, the paper does not explore whether this finding generalizes to other datasets or whether the computational overhead of determining this value outweighs the benefits of the network's scalability. This raises questions about whether the MTS network is genuinely easier to deploy compared to other frameworks that do not require such fine-tuning and suffer minimal performance loss compared to MTS. Including computational cost comparisons for hyperparameter tuning across datasets would provide valuable insights into the practical viability of the approach.

**Questions:**

- Given that the optimal number of layers for the MTS network was analyzed only for the Weibo dataset, what is the computational cost of determining this parameter on other datasets? Could this tuning requirement offset the claimed ease of deployment, especially compared to frameworks that perform well with fixed configurations?

- Has the MTS network been evaluated on real-world internet datasets that include diverse languages, modalities, and sources? If not, how might the network’s reliance on Taylor series for modeling cross-modal interactions affect its robustness and performance in such heterogeneous and potentially noisy environments?

**Reviewer Confidence:**

2: The reviewer is willing to defend the evaluation, but it is likely that the reviewer did not understand parts of the paper

**Scope:**

4: The work is relevant to the Web and to the track, and is of broad interest to the community